# The Effects of Exogenous Salicylic Acid on Endogenous Phytohormone Status in *Hordeum vulgare* L. under Salt Stress

**DOI:** 10.3390/plants11050618

**Published:** 2022-02-24

**Authors:** Hülya Torun, Ondřej Novák, Jaromír Mikulík, Miroslav Strnad, Faik Ahmet Ayaz

**Affiliations:** 1Faculty of Agriculture, Düzce University, 81620 Düzce, Turkey; 2Laboratory of Growth Regulators, Faculty of Science, Palacký University and Institute of Experimental Botany, The Czech Academy of Sciences, CZ-78371 Olomouc, Czech Republic; ondrej.novak@upol.cz (O.N.); jaromir.mikulik@upol.cz (J.M.); 3Faculty of Science, Karadeniz Technical University, 61080 Trabzon, Turkey; faa@ktu.edu.tr

**Keywords:** barley, *Hordeum vulgare*, salicylic acid, salt stress, phytohormones

## Abstract

Acclimation to salt stress in plants is regulated by complex signaling pathways involving endogenous phytohormones. The signaling role of salicylic acid (SA) in regulating crosstalk between endogenous plant growth regulators’ levels was investigated in barley (*Hordeum vulgare* L. ‘Ince’; *2n* = 14) leaves and roots under salt stress. Salinity (150 and 300 mM NaCl) markedly reduced leaf relative water content (RWC), growth parameters, and leaf water potential (LWP), but increased proline levels in both vegetative organs. Exogenous SA treatment did not significantly affect salt-induced negative effects on RWC, LWP, and growth parameters but increased the leaf proline content of plants under 150 mM salt stress by 23.1%, suggesting that SA enhances the accumulation of proline, which acts as a compatible solute that helps preserve the leaf’s water status under salt stress. Changes in endogenous phytohormone levels were also investigated to identify agents that may be involved in responses to increased salinity and exogenous SA. Salt stress strongly affected endogenous cytokinin (CK) levels in both vegetative organs, increasing the concentrations of CK free bases, ribosides, and nucleotides. Indole-3-acetic acid (IAA, auxin) levels were largely unaffected by salinity alone, especially in barley leaves, but SA strongly increased IAA levels in leaves at high salt concentration and suppressed salinity-induced reductions in IAA levels in roots. Salt stress also significantly increased abscisic acid (ABA) and ethylene levels; the magnitude of this increase was reduced by treatment with exogenous SA. Both salinity and SA treatment reduced jasmonic acid (JA) levels at 300 mM NaCl but had little effect at 150 mM NaCl, especially in leaves. These results indicate that under high salinity, SA has antagonistic effects on levels of ABA, JA, ethylene, and most CKs, as well as basic morphological and physiological parameters, but has a synergistic effect on IAA, which was well exhibited by principal component analysis (PCA).

## 1. Introduction

Abiotic stresses including soil salinity, high temperatures, and a lack of fresh water present severe agricultural challenges. Soil salinization is a particularly severe problem that is promoted by intensive agronomic practices, poor water management, irrigation with inadequate drainage systems, long hot and dry seasons, and high levels of evaporation. Moreover, although the world’s population is growing steadily, roughly 793 million people receive insufficient nourishment to support an active and healthy lifestyle [1]. To overcome these problems, agriculture and food production systems must adapt to the adverse effects of climate change and become more resilient, productive, and sustainable. 

Phytohormones are natural plant growth regulators (PGRs) that act as signaling molecules and are present in plants at very low concentrations. They are key regulators of complex root to shoot interactions that control plant growth and development. Four of the known phytohormone groups—auxins, CKs, gibberellins, and brassinosteroids—are usually regarded as growth stimulators, while abscisic acid (ABA), ethylene, salicylic acid (SA), and jasmonic acid (JA) are commonly regarded as stress hormones that mediate stress responses. However, there is extensive cross-talk between phytohormone signaling pathways, so PGRs of all kinds can have additive, synergistic, or antagonistic effects on metabolic and signaling pathways [2], and can also play significant roles in plants’ responses to biotic and abiotic stresses [3]. 

The dynamics of endogenous phytohormones have been studied under conditions of drought [4,5] and salinity [6] alone and in combination [7], and also during cold acclimation [8,9,10]. However, only one published study examined endogenous phytohormone dynamics in plants subjected to an abiotic stress while also being treated with an exogenous phytohormone: Shakirova et al. investigated the mode of action of SA in salinity-stressed wheat seedlings and used immunoassays to determine its effects on the accumulation of ABA, CKs, and indole-3-acetic acid (IAA) [11]. Unfortunately, the results obtained in that work are insufficiently comprehensive to draw general conclusions about the effects of exogenous phytohormone treatment on abiotic stress tolerance and endogenous phytohormone levels. Clarifying the phytohormone signaling mechanisms that regulate the physiological aspects of salt stress responses could provide deeper insights into abiotic stress responses in general and phytohormone cross-talk, possibly revealing new ways to enhance salt stress adaptation in key crops. 

SA is an endogenous signaling molecule derived from hydroxybenzoic acid. It is classified as a hormone-like substance [12] and is predominantly active in plant immune responses to avirulent pathogens [13]. Most studies on its effects have focused on SA-induced systemic acquired resistance. However, like other phytohormones, it is also involved in regulating abiotic stress responses. The complexity of SA’s mode of action [14,15] is increased by the fact that its effects on plants depend on the duration of exposure, the studied cultivars, their compatible doses, and the concentrations of other external agents. Treatment with exogenous SA can alleviate adverse effects resulting from pathogen virulence, heavy metal stress, salt stress, and stress induced by various toxins. For instance, SA treatment has been used to counteract the detrimental effects of salt stress on key physiological, biochemical and molecular parameters in wheat [14], *Arabidopsis thaliana* (L.) Heynhold [15,16], barley [17], maize [18], *Brassica juncea* (L.) Czern. et Coss. [19], mung bean [20], and *Torreya grandis* Fortune ex Lindl. [21]. These studies focused on antioxidant mechanisms, growth inhibition, seed germination, and the biochemical composition of the SA-treated plants. However, the effects of SA treatment on the physiology and biochemistry of endogenous phytohormones and the associated molecular signal transduction mechanisms in salinity-stressed plants are unknown. 

Barley (*Hordeum vulgare* L.) is one of the most ancient and widely cultivated cereal grain crops [22]. Compared to other cereals, it adapts readily to diverse environmental conditions, and it is considered more salt-tolerant than other Triticeae members [23,24]. While barley has been used as food since the time of the Sumerians [25], it is currently mainly used in animal feed, malting, brewing, and biodiesel production [22]. However, its nutritional value is high, and it could also be an important food crop.

Therefore, the present investigation was conducted to study phytohormone levels in barley seedlings exposed to salt stress with and without exogenous SA treatment in order to determine how SA affects salt stress responses and hormonal cross-talk. Specifically, the objectives were to: (i) evaluate the effects of salt stress on the growth and water status of barley seedlings, (ii) assess the effectiveness of exogenous SA at alleviating salt stress-induced growth suppression in roots and shoots, (iii) determine how salt stress affects endogenous phytohormone levels and understand how these PGRs ameliorate the growth-inhibiting effects of salt stress, and (iv) determine how exogenous SA treatment affects endogenous phytohormone levels in the leaves and roots of barley plants under saline and non-saline conditions. To these ends, the growth parameters, leaf relative water content (RWC), leaf water potential (LWP), proline content, and levels of endogenous phytohormones including IAA, CKs, ABA, JA, and ethylene were determined in the studied barley seedlings. 

## 2. Results

### 2.1. Growth Parameters

The effects of the specified experimental conditions on the growth and physiological parameters of barley seedlings cultivated in a hydroponic system are shown in Figure 1 and Table 1. In plants not treated with SA, increasing salinity significantly (*p* < 0.01 or 0.05) and markedly reduced the seedlings’ shoot and root lengths as well as their fresh and dry weights. Specifically, relative to the C treatment, the 300 treatment reduced the lengths of roots and shoots by 50.5% and 13.6%, the fresh weights of roots and shoots by 71.9% and 48.4%, respectively, and the dry weights of roots and shoots by 37.3% and 21.6%, respectively. The 150 treatment reduced root and shoot lengths by 41.6% and 13.8% relative to C, respectively, and reduced the fresh weights of roots and shoots by 30.8% and 41.1%, respectively. However, the dry weights of roots and shoots under the 150 treatment did not differ significantly from those for control plants. Similarly, a 24 h SA pre-treatment reduced the lengths of roots and shoots by 35.6% and 10.1%, respectively, their fresh weights by 66% and 44%, respectively, and their dry weights by 49% and 44%, respectively, relative to control plants. Exogenous SA did not significantly alleviate salt-induced negative effects on growth parameters; salt-stress plants exhibited reduced root and shoot lengths independently of SA pre-treatment (Figure 1).

### 2.2. Leaf Relative Water Content (LRWC)

The 150 mM NaCl treatment did not significantly reduce the barley seedlings’ leaf RWC relative to the control, but the 300 mM NaCl treatment reduced it by 11.1% (Table 1). SA pre-treatment under non-saline conditions increased RWC by 4.3% relative to the control treatment. However, there were no significant differences in RWC between saline treatments with and without SA pre-treatment. 

### 2.3. Leaf Water Potential (LWP; Ψ_w_)

Like the RWC, the Ψ_w_ under the 150 treatment was not significantly different from that for the control treatment. However, under the 300 treatment it was 65% lower than in controls (Table 1). SA pre-treatment had little effect on Ψ_w_: the Ψ_w_ value under the SA treatment was identical to that in control plants, while that for seedlings grown in 300 mM NaCl with SA pre-treatment (i.e., under the SA300 conditions) was significantly lower than that for seedlings with SA pre-treatment under non-saline conditions. 

### 2.4. Proline

The proline concentration profiles of the roots and shoots were quite similar, as shown in Figure 2a,b. In both cases, the proline content generally increased with the salinity. This increase was substantially more pronounced in the roots. The proline contents of the leaves and roots under the 150 conditions (150 mM NaCl without SA pre-treatment) did not differ greatly from those in control plants. However, under the 300 conditions (300 mM NaCl without SA pre-treatment), the proline contents of the leaves and roots were around 2.7− and 4−fold higher than in control plants, respectively. SA pre-treatment had no significant effects on the proline content of the roots of salt-stressed plants. However, the proline content of 150 mM salt-stressed leaves in seedlings pre-treated with 0.5 mM SA was 23.1% higher than in seedlings grown at the same salt concentration without SA pre-treatment. Compared to the untreated control, SA pre-treatment under non-saline conditions increased the proline contents of leaves and roots by 17.7% and 34.2%, respectively. 

### 2.5. Endogenous Phytohormone Levels

Salicylic acid is a phenolic-like plant growth regulator (PGR). To determine the effects of exogenous SA on endogenous PGRs under salt-stress conditions, the concentrations of free auxin, CKs, ABA, JA, and ethylene were measured in the barley seedlings’ leaves and roots. The concentrations of endogenous phytohormones under the various treatments are shown in Figure 3 and Figure 4, Table 2 and Table 3, and Appendix A.

#### 2.5.1. Effects of Salinity and SA Pre-Treatment on CK Levels in Leaves 

The cytokinins considered in this study were divided into 5 groups: free bases (CK-Bs), ribosides (CK-Rs), nucleotides (CK-NTs), *O*-glucosides (CK-*O*-Gs), and 9-glucosides (CK-9-Gs). Table 2 shows the measured endogenous levels of these groups and the total cytokinin contents of leaves and roots under the tested conditions. In leaves, the moderately saline 150 conditions caused no significant changes in free base levels relative to the control treatment but slightly increased the level of CK-Rs (23.2%) and reduced the levels of CK-NTs (45.6%), CK-*O*-Gs (9.3%), and CK-9-Gs (22.5%). The total endogenous CK level was very similar to that under control conditions, indicating that cytokinin homeostasis was maintained in the leaves under the 150 conditions. However, under the highly saline 300 conditions, the levels of CK-Bs (45%), CK-Rs (2.2−fold), CK-NTs (88%), and CK-*O*-Gs (24%) in leaves were significantly higher than under control conditions, whereas those of CK-9-Gs were 31.7% lower. Additionally, the total leaf CK concentration under the 300 treatment was 42.9% higher than in control plants. Under highly saline conditions (300 mM NaCl), SA pre-treatment greatly reduced leaf CK-B and CK-R levels by 73% and 76%, respectively, relative to those seen with SA pre-treatment under non-saline conditions. Conversely, under moderately saline conditions (150 mM NaCl), SA pre-treatment increased levels of CK-Bs, CK-Rs, and CK-NTs in leaves by 44.9%, 94.9%, and 69.7%, respectively. Interestingly, levels of CK-*O*-Gs and CK-9-Gs in leaves under moderately saline conditions (150 mM NaCl) without SA pre-treatment did not differ significantly from those under control conditions. However, SA pre-treatment significantly reduced leaf CK levels under high salt-stress conditions: the levels of bioactive (CK-B) and transport (CK-R) CK forms under the SA300 conditions were 82.7% and 88.4% lower, respectively, than those under the 300 conditions. More modest reductions were observed for CK-NTs (33.8%), CK-*O*-Gs (15.9%), and CK-9-Gs (7%) when comparing the 300 and SA300 treatments (Table 2).

#### 2.5.2. Effects of Salinity and SA Pre-Treatment on CK Levels in Roots

In the absence of SA pre-treatment, saline conditions (150 mM or 300 mM NaCl) increased root levels of CK-Bs, CK-Rs, and CK-NTs relative to controls while reducing those of CK-*O*-Gs and CK-9-Gs (Table 2). Specifically, levels of CK-Bs, CK-Rs, and CK-NTs in roots under the 300 treatment were 13.4%, 45.6%, and 3.9−fold higher, respectively, under the 300 treatment than in controls; the levels of CK-*O*-Gs and CK-9-Gs were 13% and 39.5% lower, respectively. Under non-saline conditions, SA pre-treatment also increased root levels of CK-Bs, CK-Rs, and CK-NTs relative to controls while slightly reducing those of CK-*O*-Gs and CK-9-Gs. As also observed in leaves, SA pre-treatment followed by growth under saline conditions reduced levels of CK-Bs, CK-Rs, and CK-NTs in roots relative to those seen in pre-treated plants grown under non-saline conditions. Specifically, under the SA150/SA300 treatments, the levels of CK-Bs, CK-Rs, and CK-NTs were 24%/72.9%, 45.5%/92%, and 48.5%/81.7% lower, respectively, than those under the SA treatment. Conversely, SA pre-treatment increased the levels of CK-*O*-Gs by 10.9% and those of CK-9-Gs by 2.9−fold under 150 mM NaCl stress. However, under 300 mM NaCl salt stress, SA pre-treatment reduced the levels of these CKs in roots by 41.6% and 46.1%, respectively. 

#### 2.5.3. Effects of Salinity and SA Pre-Treatment on Individual CK Forms

To evaluate the effects of salt stress and exogenous SA on the distribution of specific CK types, we determined the concentrations of 6 forms of *cis*-zeatin-type CKs (*c*Z, *c*ZR, *c*Z9G, *c*ZOG, *c*ZROG, *c*ZR’5MP), 6 forms of *trans*-zeatin-type CKs (*t*Z, *t*ZR, *t*Z9G, *t*ZOG, *t*ZROG, *t*ZR’5MP), and 4 forms of isopentenyladenine-type CKs (iP, iPR, iP9G, iPR’5MP) in leaves and roots. The total measured levels of the *c*Z-, *t*Z-, and iP-type CKs in roots and leaves are shown in Table 3; results for individual CK forms are presented in Appendix A. In leaves, moderate salt stress (150 mM NaCl) caused no significant changes in the levels of *c*Z-, *t*Z-, and iP-types relative to controls. However, high salt stress (300 mM NaCl) increased the levels of *c*Z- and iP-types in leaves relative to controls; the increase for iP-types (2.4−fold) was markedly greater than that for *c*Z-types (45.7%). Conversely, levels of *t*Z-types fell by 48.5% relative to controls under these conditions. In roots, salt stress did not significantly change the levels of *t*Z- and iP-type CKs relative to controls, but modest effects on the levels of *c*Z-types were observed at 150 mM NaCl. SA pre-treatment did not significantly alter the levels of *t*Z-type CKs in leaves, independently of salt stress. However, SA pre-treatment followed by 150 mM NaCl salt stress increased levels of *c*Z- and iP-types by 16.5% and 2.4−fold, respectively, relative to 150 mM NaCl alone. Additionally, SA pre-treatment followed by 300 mM NaCl salt stress reduced the levels of these forms by 37.1% and 88.3%, respectively, relative to 300 mM NaCl alone. Under the SA300 conditions, levels of *c*Z-, *t*Z-, and iP-type CKs in roots were significantly (50%, 43.8%, and 81.3%, respectively) lower than those seen under the 300 conditions.

#### 2.5.4. Effects of Salinity and SA Pre-Treatment on IAA Levels

In the absence of SA pre-treatment, moderately saline conditions increased IAA levels in leaves but reduced them in roots: the IAA levels in the leaves under the 150 treatment were 3.3−fold higher than in control plants, but those in the roots were 66.7% lower than in controls (Figure 3a,b). However, under the highly saline 300 conditions, the IAA levels in leaves and roots did not differ significantly (*p* < 0.05) from those in control plants. SA pre-treatment at high salinity sharply increased leaf IAA levels: under the SA300 conditions, the leaf IAA level was approximately 13.7-−fold higher than in plants pre-treated with SA under non-saline conditions, and 8.4−fold higher than in plants grown in 300 mM NaCl without SA pre-treatment. Additionally, SA pre-treatment strongly alleviated salt-induced reductions in IAA levels in barley roots grown under moderate salinity: the IAA content under the SA150 treatment was 81% higher than under the 150 treatment. SA pre-treatment under non-saline conditions reduced the IAA content of barley roots by 29.1% relative to the control treatment.

#### 2.5.5. Effects of Salinity and SA Pre-Treatment on ABA Levels

ABA is regarded as a stress hormone and plays a key role in signaling pathways regulating water deficit and stomatal closure [26]. As shown in Figure 3c,d, ABA levels in leaves were much higher than those in roots. Mild salt stress (150 mM) increased leaf ABA levels 3.7−fold relative to controls. However, ABA levels in roots under 150 mM salt stress did not differ significantly from those under control conditions. High salt stress (300 mM NaCl) sharply increased ABA levels in both leaves and roots (by 33% and 5.8−fold, respectively) relative to controls. SA pre-treatment (0.5 mM) alone did not significantly change ABA levels in leaves or roots relative to controls. However, SA pre-treatment followed by growth under moderate salt stress (150 mM NaCl) increased ABA levels by 88.5% and 7.8−fold in leaves and roots, respectively, relative to those seen for the same salt stress without SA pre-treatment. Under high salt stress (300 mM NaCl), SA pre-treatment reduced ABA levels in leaves and roots by 83.5% and 69.1% relative to those seen at the same salt concentration without exogenous SA. 

#### 2.5.6. Effects of Salinity and SA Pre-Treatment on JA Levels

Pre-treatment with exogenous SA significantly affected endogenous JA levels (Figure 3e,f). In the absence of salt stress, SA pre-treatment (0.5 mM) reduced JA levels in leaves and roots by 15.1% and approximately 2−fold, respectively. Furthermore, SA pre-treatment reduced JA levels in leaves by 20.1% and 76.1% at NaCl concentrations of 150 and 300 mM, respectively, relative to the SA-free treatments at the same NaCl concentrations. Conversely, in roots, SA pre-treatment reduced JA levels (by 86%) under moderate salt stress (150 mM NaCl) but had no significant effect under high salt stress (300 mM NaCl). High salt stress without SA pre-treatment strongly reduced JA levels in roots (by 84%) relative to the non-saline control treatment. In leaves, moderate salt stress (150 mM NaCl) increased JA levels by 12.5% relative to the control treatment, but high salt stress (300 mM NaCl) caused no significant change in JA levels. 

#### 2.5.7. Effects of Salinity and SA Pre-Treatment on Ethylene Levels

Ethylene is a gaseous phytohormone, so its accumulation in barley seedlings was measured using GC-FID after placing seedlings in plastic bags to trap their gaseous emissions. Ethylene production in the barley seedlings increased with salinity (Figure 4): moderate (150 mM) and high (300 mM) salt stress increased ethylene levels by 23.9% and 63.8%, respectively, relative to the control treatment. SA pre-treatment alone increased ethylene production by 26.4% relative to the control treatment. Ethylene levels in SA pre-treated plants under moderate salt stress (150 mM) were 19% higher than in plants grown at the same salt concentration without exogenous SA. Conversely, ethylene levels in SA pre-treated seedlings under high salt stress (300 mM) were 26.4% lower than those in seedlings grown at the same salt concentration without SA pre-treatment. 

### 2.6. Principal Component Analysis

Principal Component Analysis (PCA) was used to evaluate the correlations between treatments, growth parameters, and endogenous phytohormone levels in barley seedlings. The first three components of a PCA can be considered to adequately explain the variability between samples if they collectively explain at least some predefined proportion (typically 70% or more) or the total variance in the data [27]. 

The PCA of the growth parameters revealed that the first two principal components (PC1 and PC2) explained approximately 90.82% of the total variance; PC1 explained 69.05% of the total and PC2 explained 21.76% (Figure 5). The analysis indicated that the leaf relative water content and leaf water potential (LRWC and LWP; *r* = 0.968, *p* < 0.05, Table 3), the lengths of the roots and shoots (RL and SL), and the fresh and dry weights of the roots and shoots (RFW, SFW, RDW, and SDW; range; *r* = 0.844 to 973, avg. 0.919) were positively and significantly correlated (*p* < 0.01 or 0.05). The RWC and LWP correlated strongly with the 150 mM NaCl, SA, and SA150 treatments in the upper quadrant, but all other parameters grouped with the control treatment in the right lower quadrant. The leaf and root proline contents (LPC and RPC) correlated strongly with the 300 and SA300 treatments in the left lower quadrant (range; *r* = −0.898 to −0.971). The PCA plot also shows that both the proline contents (LPC and RPC) and the leaf’s relative water content and water potential (LRWC and LWP) correlated strongly and negatively (range; *r* = −0.898 to −0.970, *p* < 0.01 or 0.05, see also the correlation matrix shown in Figure 5). 

To better visualize possible differences between the stress treatments and characterize their correlations with endogenous phytohormone levels, separate PCAs were performed based on the measured levels of the 16 CK metabolites and those of IAA, ABA, and JA, and basic morphological and physiological parameters in leaves of the barley seedlings under the various experimental conditions (Figure 6a). Accordingly, the PCA based on the measurements in the leaf yielded two principal components that collectively explained 69.84% of the total variance, with PC1 explaining 45.65% of the total (Figure 6a). In the bi−plot for this PCA, the SA + 150 mM NaCl (LSA150) treatment and control (LC) outcomes for leaves was associated and significantly strong correlated with *t*ZR (ck2), *t*Z9G (ck3), *t*ZROG (ck5), *t*ZR’5MP (ck6), *c*Z (ck7), *c*ZR (ck8), *c*Z9G (ck9) and iP (ck13) together with leaf/shoot fresh and dry weights or its length or JA (range; *r* = 0.832–0.998, *p* < 0.05) in the upper quadrant on PC1 (positive side), while remaining iPR’5MP (ck16) were not correlated with any of the morphological. There was possible association among *t*Z (ck1), *t*ZOG (ck4), iP9G (ck15), leaf water potential (LWP) and relative water content (LRWC) with treatment L150 (50 mM NaCI) in the lower quadrant on PC1 with no correlation, except LWP and LRWC within. However, *c*ZOG (ck10), *c*ZROG (ck11), *c*ZR’5MP (ck12) and iPR (ck14) were associated with L300 (300 mM NaCl) in the upper quadrant on PC2 and significantly strong correlated with LPC (*r* = 0.936–0975, *p* < 0.05) in the upper quadrant on PC2. It was noted that only IAA in the lower quadrant on PC2 was associated with LSA300 and LSA (0.5 mM SA + 300 mM NaCl and 0.5 mM SA), but not correlated. In addition, ABA level on PC2 (upper quadrant) was associated with 300 mM NaCl (L300) and significantly strong correlated with only *c*ZR′5MP (ck12, *r* = 0.882, *p* < 0.05) (Figure 6a). Five factors (F1–5) were identified for the measured parameters and hormone levels in the leaf/shoot (Table 4). Factor 1 formed the largest positive association with LC (5.892) and negative associations with LSA300 (−4.587) and L300 (−3.231), followed by F2 (L300; 4.083 LSA; −3.309), F3 (LSA300; −3.580 and LC; −2.473) and the remaining two factors with low associations (Table 4).

The first two principal components of the PCA based on the endogenous 16 CK metabolites, phytohormone levels and IAA, ABA, and JA together with basic morphological and physiological parameters in roots of the barley seedlings under the various experimental conditions explained 62.85% of the total variance in the data (Figure 6b). In the bi−plot, control (RC) outcomes for root correlate strongly (range; *r* = 0.817–0.987, *p* < 0.05) with five CK metabolites [(*t*Z (ck1), *c*ZR (ck8), *c*ZOG (ck10), *c*ZROG (ck11) and iP9G (ck15)] together with root fresh (RFW) and dry (RDW) weights in the upper quadrant. The 150 mM NaCl (R150) and 0.5 mM SA (RSA) with nine CK metabolites [(*c*ZR, *t*ZROG, *t*ZR, *c*Z, iPR, *t*ZR’5MP, iP and *c*ZR’5MP, iPR’5MP; ck2, 5−8, 12−14, and 16 respectively)] and JA in the lower quadrant on PC1 were correlated strong at significant level (range; *r* = 0.817–0.987, *p* < 0.05), except with ck5 and ck7. It was noted that JA was strong correlated with ck1, 2, 6 and 16 (*r* = 0.825–0.923, *p* <0.05). In addition, RSA150 (0.5 mM SA + 150 mM NaCI) treatment in the upper quadrant, RSA300 and R300 (0.5 mM SA + 300 mM NaCl and 300 mM NaCl) treatments in the lower quadrant on PC2 (21.84% variance) were associated but not correlated with IAA and ABA, except the strong correlation (*r* = 0.921, *p* < 0.05) between *t*ZOG (ck4) and *c*Z9G (ck9) with *t*ZOG (ck4) and *c*Z9G (ck9) in the upper quadrant on PC2. Five factors (F1–5) were identified for the measured parameters and hormone levels in the root (Table 4). Factor 1 formed the largest negative association with RSA300 (−5.372) treatment, followed by the largest positive associations with RSA (4.166) and R150 (2.671) treatments. Remaining four factors (F2−5) had positive largest associations with RC (F2; 4.343), R300 (F3; 3.835), and RSA150 (F4; 2.574), etc., or low negative associations (Table 4).

Apart from PCA, cluster analysis was also performed based on the measured parameters. The clustering dendrograms of UPGMA (unweighted pair group method with arithmetic mean) represents dissimilarity for the hormones, basic morphological and physiological parameters in leaves/shoots (Figure 7a) and roots (Figure 7b). As shown in the figure, the average values of the measurements yielded 3 main clusters for the leaf/shoot and the root each. The third and first cluster in Figure 7a and the third and second cluster in Figure 7b had relatively high levels of endogenous hormone levels and values of basic morphological and physiological parameter measures that they significantly or insignificantly induced or reduced by the exogenous SA under salt-stress conditions.

## 3. Discussion

Salt stress adversely affects plant growth and development via various molecular, physiological, and biochemical processes. It has therefore been extensively studied to clarify the mechanisms of plant adaptation to unfavorable environmental conditions. Salinity tolerance mechanisms are mainly controlled by natural plant growth regulators (PGRs) such as salicylic acid (SA) [28]. This study was designed to shed new light on the mechanism of plant adaptation to salt stress by measuring changes in endogenous phytohormone levels in barley seedlings and their responses to treatment with exogenous SA at three salt concentrations (0, 150 mM, and 300 mM NaCl, corresponding to no salt stress, moderate salt stress, and high salt stress, respectively). The results obtained provide new insights into endogenous phytohormone crosstalk under salt stress, the physiological responses underpinning salinity tolerance, and the potential to enhance salinity tolerance in crops by treatment with exogenous PGRs. 

Reducing growth is typically among the first plant responses to severe environmental stress. In this work, increasing salinity significantly reduced growth, particularly in the roots. As the salt in our study was applied in Hoagland solution, the salt stress was first perceived in the roots, which were therefore also the first organ to exhibit a physiological stress response [29]. Consequently, the responses of roots and shoots to salt stress were different. Although moderate salt stress (150 mM NaCl) dramatically reduced growth parameters (other than dry weight) in both vegetative organs, the length, dry weight, and fresh weight of shoots under moderate salt stress did not differ significantly from those under high salt stress (300 mM NaCl). Moreover, necrotic symptoms in leaves were not observed morphologically in this work. SA treatment alone also had significant reductions on the growth parameters of barley seedlings. Contrary to our results, there were no significant differences in *Dianthus superbus* growth or physiological responses between plants treated with exogenous SA and untreated controls [30]. 

It is well known that barley is more salt tolerant than other Triticeae members [31]. In this work, barley seedlings under high salt stress (300 mM NaCl) exhibited acute symptoms, particularly with respect to their leaf water status. Relative water content (RWC) is an important indicator of plant water status under stress conditions and usually declines as salinity increases. Interestingly, under saline conditions, SA pre-treatment had no significant effect on barley leaf RWC as compared to non-SA-treated salt-stressed plants. This outcome is inconsistent with the results of Li et al. [21] and El Tayeb [17], who found that SA treatment slightly increased leaf RWC in salt-stressed *Torreya grandis* [21] and barley [17] plants. Our results also showed that plants exposed to 300 mM NaCl had lower leaf water potential (LWP) values than those grown at other salt concentrations. The trends in LWP mirrored those in RWC, especially in seedlings subject to high salt stress (300 mM NaCl). Notably, even under the most strongly saline conditions tested, the seedlings’ RWC values remained above 77%. *H. vulgare* ‘Ince’ can thus maintain its water content even under severe salt stress. The RWC and the fresh and dry weights of leaves and roots decreased gradually with increasing salt stress, while the proline content increased. In keeping with these findings, salinity caused pronounced changes in growth parameters, RWC, and proline levels in wheat [11], barley [17], *Torreya grandis* [21], and soybean [32]. SA pre-treatment did not significantly alter these physiological parameters in salt-stressed plants. However, it did non-significantly increase RWC under high salt stress, presumably because of osmotic changes resulting from high proline accumulation (see below), which can help maintain leaf turgor under saline conditions. Proline may also help regulate LWP under salt stress [33,34,35]. 

Severe environmental conditions cause the accumulation of osmoprotectants that ensure continuity of plant cell growth and development. Proline is a compatible solute for osmotic adjustment that accumulates in large quantities in response to biotic and abiotic environmental stresses [36]. It also acts as a chelating agent and has signaling functions and antioxidant activity that are important in plants [37]. Accordingly, the proline content of the barley seedlings’ leaves and roots increased with the salinity. SA pre-treatment increased proline levels in leaves and roots under non-saline conditions but had no significant effect on the proline content of barley roots under saline conditions. However, SA pre-treatment did induce proline accumulation in barley leaves exposed to 150 mM NaCl, probably reflecting a mechanism of adaptation to salinity-induced osmotic stress. A similar study on wheat found that seedlings pre-treated with SA exhibited increased salinity tolerance that was attributed to elevated proline levels [11]. Similar results have been obtained in lentils pre-treated with 0.5 mM SA [35] and in *Torreya grandis* [21]. These findings suggest that proline is an important component of SA-induced defense reactions to salt stress in barley: treatment with exogenous SA ameliorates the adverse effects of salt stress by promoting proline accumulation. Under highly saline conditions (300 mM NaCl), SA pre-treatment did not increase proline accumulation, possibly because of the accumulation of other osmolytes. It would be desirable in future to identify these other osmolytes and determine how their intracellular concentrations vary with salinity. On the other hand, in our previous study with other barley cultivars grown under salt stress, positive effects of SA were observed even at high salt-stress concentration [24]. However, contrary to our study, the negative effects of SA on barley cultivar (Ince-04) were determined under salt stress in this observation. In view of these reports, as well as the data reported here, it might be evident that effective concentration of SA vary from species to species as well as in cultivars belonging to the same species. Phytohormones are critical regulators of plant growth and development. Since changes in hormone levels are believed to be essential for controlling growth under environmental stress, it is important to understand how their levels vary under stress conditions and the crosstalk that exists between phytohormone signaling pathways. We therefore investigated the effects of exogenous SA treatment on endogenous phytohormone levels in barley seedlings under normal and saline conditions. To this end, we measured the levels of free auxin, CKs, ABA, JA, and ethylene, all of which are key natural plant growth regulators (PGRs). 

IAA is a growth-promoting hormone that plays a major signaling role in plants. At 150 mM NaCl, its concentration in roots was much lower than under the control treatment, whereas that in the leaves was higher than in controls. Conversely, at 300 mM NaCl, its concentrations in roots and shoots did not differ significantly from those under control conditions. Several previous studies have suggested that auxin is involved in salt stress responses in plants, but little is known about how these responses are regulated [38]. The IAA levels observed in salt-stressed barley roots in this work are consistent with the values reported by Dunlop and Binzel [39]. However, in contrast to our results, salt stress sharply reduced IAA levels in leaves of *Iris hexagona* [40], tomato [6], and wheat [11]. This may be related to the simultaneous leaf and root growth suppression seen in these studies. The observed increase in IAA levels in barley leaves under salt stress may facilitate the relocation of nutrients away from dying leaves by delaying leaf abscission. However, the increased accumulation of IAA in leaves at 150 mM NaCl may not allow root growth to be maintained. SA pre-treatment under non-saline conditions increased IAA levels in roots but not in leaves. Similarly, SA pre-treatment had no significant effect on IAA levels in the leaves of barley seedlings under moderate salt stress (150 mM NaCl) but increased them in roots compared to 150 controls. Conversely, the SA + 300 mM NaCl treatment dramatically increased IAA levels in leaves but had no significant effect on those in roots. However, a study on wheat revealed that SA treatment prior to sowing prevented an NaCl-induced decline in IAA levels of 2% [11]. 

Like auxins, CKs regulate several processes of plant growth and development. The levels of cytokinin-type hormones are good indicators of plants’ environmental stress resistance [41] because CKs generally enhance tolerance of abiotic stresses such as high salinity and high temperatures [42]. For example, increasing CK levels were found to improve cereals’ resistance to salt stress [43]. In this work, the levels of 16 different CK metabolites were determined in both leaves and roots of barley seedlings grown under the various experimental conditions. Total CK concentrations were lower in roots than in leaves under salt stress. This is consistent with the fact that some CKs synthesized in the roots are translocated to the shoots and thus influence shoot responses. Accordingly, the effects of salt stress on shoot growth were less pronounced than those on root growth. Many studies have shown that salt stress can reduce overall CK levels [11,43]. However, we observed a strong increase in total CK levels in salt-stressed barley leaves and roots: levels of all cytokinin metabolite groups other than CK ***O***- and 9-glucosides increased. We believe that this effect is probably an early response to salt stress. When the levels of individual CK metabolites were measured after 4 days of salt stress, significant increases in the abundance of bioactive CK bases, CK ribosides (transport forms), and CK nucleotides were detected in the barley seedlings’ leaves and the roots. These high levels of bioactive CKs may enhance the growth of salt-stressed barley. Since most studies in this area have found that salt stress reduces levels of endogenous CKs [44], the observed increase in CK-Bs, CK-Rbs, and CK-Ntds is rather unusual and may well be genotype-dependent. Therefore, deeper studies are needed to explore the regulation of endogenous CK profiles in salt-tolerant/resistant barley cultivars grown under saline conditions. Treatment with exogenous SA yielded more complex results that depended strongly on the salt concentration. As in wheat [11], SA pre-treatment induced no significant changes in the CK levels of barley leaves and roots under non-saline conditions. At a moderate NaCl concentration (150 mM), SA pre-treatment increased CK levels, especially in leaves. Conversely, at 300 mM NaCl, SA pre-treatment dramatically reduced levels of all CK groups in barley roots as well as those of CK-Bs, CK-Rs, and CK-9-Gs in leaves; levels of CK-NTs and CK-*O*-Gs in leaves were also reduced, but to a lesser extent. Therefore, increasing salinity has opposing effects on the accumulation of IAA and CKs. These results suggest that SA acts as a CK antagonist under highly saline conditions. 

Small decreases in LWP are thought to slow down ABA metabolism [45]. ABA regulates stomatal closure and is one of the most important stress indicators and stress response mediators in plants. Therefore, salt stress increases its concentration in leaves, as shown in Figure 3c: increases in salinity caused strong increases in ABA levels. In keeping with these findings, endogenous ABA levels in the leaves of heat- and drought-treated tobacco plants were much higher than those in the leaves [7]. Several other publications have also reported increases in ABA levels induced by adverse environmental conditions [6,7,11,46]. Under non-saline conditions, SA pre-treatment had no effect on endogenous ABA levels. Conversely, under moderately saline conditions (150 mM NaCl), SA pre-treatment strongly increased ABA levels in both vegetative organs; the increase was more pronounced in roots (7.8−fold) than in leaves (88.5%). This may be due to SA-induced pre-adaptation of the barley seedlings. In accordance with an earlier study on wheat [11], SA pre-treatment under salt stress promoted the accumulation of proline and ABA in leaves. However, it should be noted that leaf ABA levels under the SA300 conditions (SA pre-treatment followed by growth at 300 mM NaCl) were lower than those for SA pre-treatment followed by growth under non-saline conditions. 

There have been few studies on changes in endogenous JA levels under salt stress. In this work, the JA levels in the leaves were higher than in the roots. Without SA pre-treatment, moderate salt stress (150 mM NaCl) increased JA levels in barley leaves and roots but severe salt stress (300 mM NaCl) strongly reduced JA levels in both vegetative organs. Previous studies on tomato [47] and rice [48] indicated that salt stress increased endogenous JA levels. In addition, Pedranzani et al. [47] stated that the increased JA levels observed in salt-tolerant cultivars under saline conditions may contribute to their salt tolerance. These results and those presented here suggest that JA accumulation may be a component of mechanisms that protect against salt stress. Previous studies on JA and SA in plants have mainly focused on the effects of biotic stress [49,50], although there have also been some investigations into their roles in responses to abiotic stresses [51,52]. In our study, under non-saline conditions, SA pre-treatment reduced JA levels in leaves but significantly increased them in roots. Conversely, under saline conditions, SA pre-treatment strongly reduced JA levels in both vegetative organs. This indicates that there is an antagonistic relationship between SA and JA under salt stress. In agreement with our results, Riemann et al. [52] reported a negative crosstalk between SA and JA signaling. Our results also showed that the effects of SA pre-treatment followed by high salt stress (300 mM NaCl) on JA levels were similar to those on CKs. These findings suggest that there is an antagonistic relationship between JA and SA, as previously shown in Arabidopsis [53] and flax [54].

Like SA and JA, ethylene is a low molecular weight signaling molecule [55] involved in defense regulation under environmental stress conditions. In our study, salinity increased ethylene levels in barley seedlings. SA pre-treatment also increased ethylene levels under non-saline and moderately saline conditions but reduced them under highly saline conditions. This is consistent with the report of Tirani et al. [56], who found that there was an antagonistic relationship between SA and ethylene in canola plants. Many studies have shown that jasmonate and ethylene act synergistically in opposition to salicylic and abscisic acid [55,57,58]. In our study, SA increased levels of ABA and ethylene at moderate salt concentrations (150 mM NaCl) but reduced them at high salt concentrations (300 mM NaCl). Additionally, SA pre-treatment reduced JA levels at both NaCl concentrations. SA thus has antagonistic effects on ABA, JA, and ethylene levels under high salt stress. 

The differential responses of exogenous SA application under salt-stress conditions extracted several principal components (PCs) depending on the measurements in leaf/shoot and root in the barley seedlings. As can be seen from the PCs, in the leaf, 0.5 mM SA + 150 mM NaCl (LSA150) treatment significantly increased levels of *c*Z (ck7) and *c*ZR (ck8) compared to 150 mM NaCl (L150) treatment, and 300 mM NaCl (L300) treatment increased levels of ck10 and ck14 compared to 0.5 mM SA + 300 mM NaCl (LSA300) in comparison to their controls (LC and LSA). In the root, 0.5 mM SA + 150 mM NaCl (RSA150) treatment compared to 150 mM NaCl (R150), and 300 mM NaCl and 0.5 mM SA + 300 mM NaCl (RSA300) treatment significantly promoted the increase of the amount and levels of the measured parameters in comparison to their controls (RC and RSA). The PCs also well exhibited the endogenous ABA increase in the leaf and root under severe salinity (300 mM NaCI) condition and its reduction when SA combined with the salinity (300 mM); 150 mM NaCl increased the level of IAA, decreased it when combined with SA, but increased it with 300 mM NaCl treatment when combined with 0.5 mM SA in comparison to controls and other treatments. In the root, however, a dose-dependent increase in IAA level at 150 and 300 mM NaCl treatment was maintained, even when these salt concentrations were combined with SA, but more so than the controls. These PCs also confirm stress alleviation effect of exogenous SA in plants in salt-stress example in the present study.

As phytohormones are important signaling molecules, their accumulation plays important roles in regulating growth as well as developmental and physiological processes associated with stress responses. This study investigated changes in the levels of endogenous phytohormones induced by SA pre-treatment under salt stress. In barley, salt stress led to proline accumulation while reducing the RWC and LWP of leaves and roots, and lowering their levels of IAA, total CKs, ABA, and JA. Exogenous SA treatment thus causes pronounced changes in endogenous phytohormone levels in barley seedlings under various salt-stress conditions. These results also show the combined effects of exogenous SA and salt stress on phytohormone levels. The varied changes in the accumulation and metabolism of endogenous phytohormones induced by SA pre-treatment under salt-stressed barley suggest that there is synergistic and antagonistic crosstalk between PGRs, which presumably negatively supports the maintenance of plant growth and development.

## 4. Materials and Methods

### 4.1. Plant Material and Salt-Stress Conditions

Seeds of the barley cultivar *Hordeum vulgare* L. ‘Ince-04′; Poaceae; *2n* = 14 obtained from the Transitional Zone Agricultural Research Institute (Eskişehir, Turkey) were used in this work. Barley seeds were surface-sterilized with 70% ethanol for 5 min, rinsed with sterile deionized water, and then immersed in 5% sodium hypochlorite solution for 15 min. To remove this solution, seeds were washed at least five times with sterile deionized water. After sterilization, seeds were sown in rock wool plugs soaked with half-strength Hoagland’s nutrient solution (one seed per plug) [59]. The plugs were placed in deep pots holding 16 plugs each, and the plants were grown in an aerated hydroponic system containing Hoagland’s solution, which was renewed every two days. The pots were placed in a growth chamber with a 16/8 h photoperiod, day/night temperatures of 22/18 °C, an irradiance of 300 μmol m^−2^ s^−1^, and 70/65% humidity. After 16 days under these conditions, pots were randomly divided into six experimental groups: C: untreated control plants, 150: plants treated with 150 mM NaCl to induce moderate salt stress, 300: plants treated with 300 mM NaCl to induce high salt stress, SA: plants treated with 0.5 mM SA for 24 h (0 mM NaCl), SA150: plants preincubated for 24 h with 0.5 mM SA then cultivated for 4 days in 150 mM NaCl, SA300: 0.5 mM SA pre-treatment for 24 h followed by growth in 300 mM NaCl for 4 days. SA and/or NaCl were applied to the plants in a Hoagland nutrient solution. There were 2 replicates per treatment and 16 plants per replicate. At the end of the 4 day-NaCl treatment period, the barley plants were harvested. Roots and shoots were harvested separately from all groups of seedlings and stored at −80 °C for further analysis. 

### 4.2. Growth Measurements

Seven seedlings of each group in the experimental design were taken randomly and separated into shoots and roots. Shoot and root lengths were determined by measuring the average length of the longest leaves and roots, respectively. The fresh weights (FW) of the leaves and roots were determined by weighing, then the samples were dried at 80 °C for 48 h and weighed again to determine their dry weight (DW). 

### 4.3. Leaf Relative Water Content

After harvest, six leaves from each group were weighed to determine their fresh weight (FW). The leaves were then floated on deionized water at 4 °C for 16 h, after which the weights of the turgid leaves (TW) were measured. The turgid leaves were dried in an oven at 70 °C for 48 h, after which their DW was measured. Finally, the leaves’ relative water content (RWC) was calculated using the following equation:RWC (%) = (FW−DW/TW−DW) × 100

### 4.4. Leaf Water Potential

The leaf water potential (Ψ_w_) of the barley leaves was measured using a PSYPRO water potential system (Wescor, Logan, UT, USA) equipped with a thermocouple psychrometer chamber after 60 min of equilibration. Measurements were conducted until at least three consistent measurements were obtained for barley leaves representing each individual experimental group.

### 4.5. Proline Content

The free proline content was measured as described previously [60]. Leaf and root samples were homogenized in 3% sulphosalicylic acid and the homogenate was filtered through Whatman’s No. 2 filter paper. Extracts were assayed for proline using the acid-ninhydrin method, in which the proline content (μmol proline g^−1^ FW) is determined by UV-VIS spectrophotometry (Thermo, Evolution 100, UK) using a standard curve based on proline solutions of known concentration. 

### 4.6. Phytohormone Analysis

The concentrations of endogenous phytohormones (auxin, cytokinins, ABA, JA, and ethylene) in the leaves and roots of the barley cultivar ‘Ince’ were determined by ultra-performance liquid chromatography–electrospray tandem mass spectrometry (UHPLC–MS/MS). All measurements were done in triplicate. 

Briefly, free IAA was purified from leaves and roots, and quantified as described by Pěnčík et al. [61]. Separation was performed using an ultra-high performance liquid chromatograph (Acquity UPLC; Waters; Milford, MA, USA) equipped with a Symmetry C18 column (5 µm, 2.1 × 150 mm; Waters), and the effluent was introduced into the electrospray ion source of a Quatro micro API tandem quadrupole mass spectrometer (Waters; Milford, MA, USA). 

Cytokinin extraction and purification was performed as described by Novak et al. [62], and cytokinin levels were quantified by ultra-high performance liquid chromatography–electrospray tandem mass spectrometry (UHPLC–MS/MS) [63]. 

Endogenous ABA was isolated by solid-phase extraction using Oasis HLB cartridges (60 mg, 3 mL; Waters) and quantified using a UHPLC–MS/MS system [64]. 

Endogenous JA was extracted after overnight extraction with 80% (*v*/*v*) methanol and its concentration was quantified by UHPLC–MS/MS [65]. In the UHPLC-MS/MS experiments, samples were analyzed in MRM mode using optimized cone voltages and collision energies for diagnosis of each phytohormone. Stable isotope-labeled internal standards were used as references to quantify levels of the target analytes. Data analysis was performed using Masslynx™ 4.1 (Waters, Milford, MA, USA) and the phytohormones were quantified by the standard isotope-dilution method using three technical replicates per biological sample.

Finally, the gas chromatography-flame ionization detector (GC-FID) method [66] was used to determine endogenous ethylene concentrations. Intact barley plants from each experimental group were placed in separate gas-tight plastic bags containing 10 mL of the appropriate growth medium. The bags were then tightly sealed with rubber septa and the plants were grown in the light for 1 h. Air (1 mL) was then removed from each bag using a syringe and analyzed using a Finnigan Trace GC Ultra equipped with a FID detector and 50 m capillary column (HP-AL/S stationary phase, 15 μm, i.d. = 0.535).

### 4.7. Statistical Analysis

All analyses were done using a completely randomized design. Two biological replicates with three technical replicates each (*n* = 6) were performed for each experiment. All data were subjected to one-way analysis of variance (ANOVA) and the significance of differences between treatments was evaluated using Duncan’s multiple range test with a significance threshold of *p* < 0.05. A statistical software package was also used to do principal components analysis (PCA) (Addinsoft 2019, XLSTAT and Data Analysis Solution, Version 2019.3.2., New York, NY, USA). The correlation coefficients were determined for the endogenous hormone levels and values of basic morphological and physiological measurements in barley seedlings under salt stress using exogenous SA applications. To achieve a better understanding of the similarities and differences in the profiles’ hormones, and basic morphological and physiological measurements in leaf/shoot and root in the seedlings, using Agglomerative Hierarchical Clustering (AHC) in XLSTAT, was used with the data set of contents and measurement as variables. Grouping of stress treatments with control was done using cluster analysis (UPGMA, dissimilarity, standardized observations) using XLSTAT.

## 5. Conclusions

We interpreted the interactions between exogenous SA and endogenous phytohormones in terms of possible synergistic and antagonistic crosstalk and effects under salt- stress conditions. Barley seedlings exposed to NaCl (150 and 300 mM) in the hydroponic system exhibited rapid reductions in growth parameters, LWP, and LRWC, together with strong accumulation of the stress indicator ABA. In the absence of SA pre-treatment, salt stress increased the levels of total CKs, ABA, JA, and ethylene in leaves and roots, while reducing those of IAA. Under non-saline conditions, SA pre-treatment did not significantly change the levels of endogenous phytohormones but did markedly alter plant-water relations and increase proline accumulation. SA pre-treatment appeared to enhance salt tolerance by modulating the accumulation of all the studied PGRs in both roots and shoots. Specifically, in barley plants subject to high salt stress, SA appeared to have antagonistic effects on JA, ABA, and CKs, and ethylene, as well as a synergistic effect on IAA levels. 

## Figures and Tables

**Figure 1 plants-11-00618-f001:**
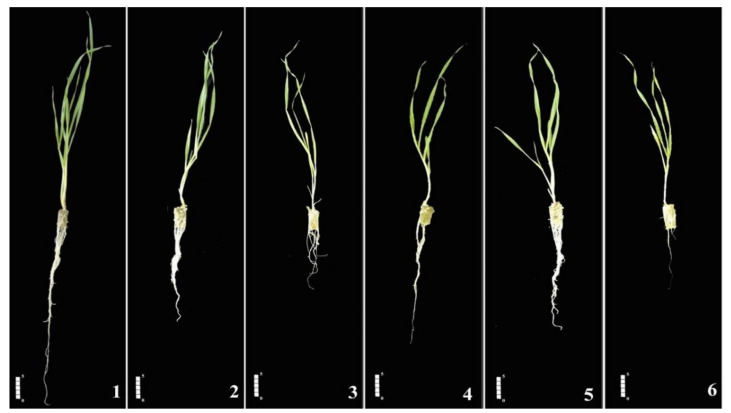
Effect of salicylic acid (SA) pre-treatment on growth of *Hordeum vulgare* L. ‘Ince-04′ plants under control and saline conditions (150 and 300 mM NaCl): (**1**) Control; (**2**) 150 mM NaCl; (**3**) 300 mM NaCl; (**4**) 0.5 mM SA pre-treatment without NaCl stress; (**5**) 0.5 mM SA pre-treatment with 150 mM NaCl; (**6**) 0.5 mM SA pre-treatment with 300 mM NaCl. For SA pre-treatment, plants were irrigated with 0.5 mM SA for 24 h in a hydroponic system. Scale bar, 5 cm.

**Figure 2 plants-11-00618-f002:**
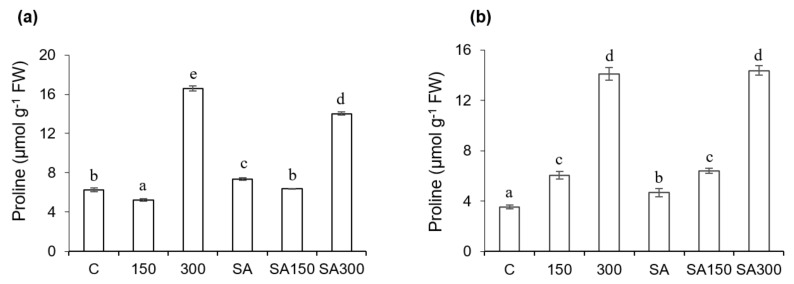
Effect of salicylic acid (SA) pre-treatment on proline content of *Hordeum vulgare* L. ‘Ince-04′ plants. Leaves (**a**) and roots (**b**) under control and saline conditions (150 and 300 mM NaCl). Different letters indicate significant differences between treatments (*p* < 0.05).

**Figure 3 plants-11-00618-f003:**
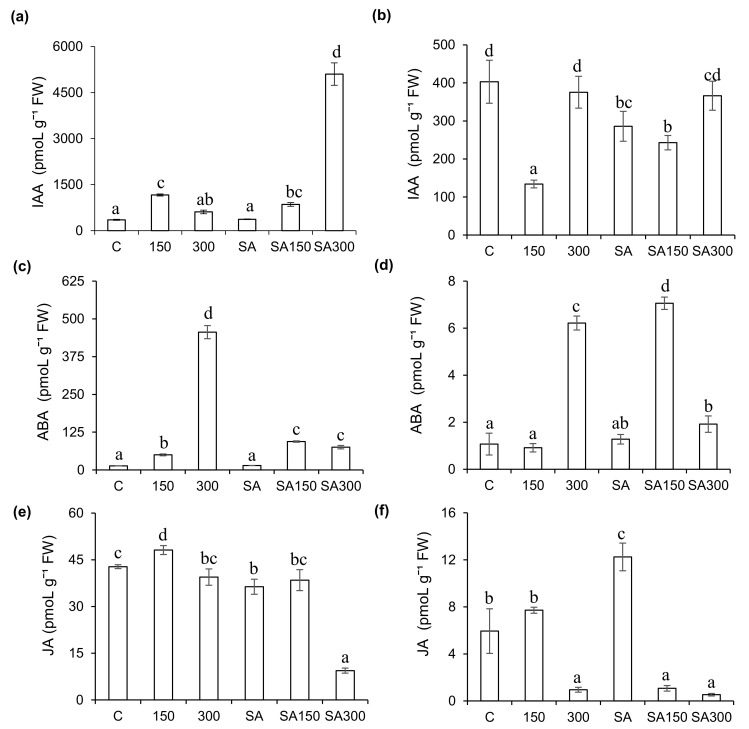
Effect of salicylic acid (SA) pre-treatment on indol-3-acetic acid (IAA), abscisic acid (ABA), and jasmonic acid (JA) levels of *Hordeum vulgare* L. ‘Ince-04′ plants leaves. Leaves (**a**,**c**,**e**) and roots (**b**,**d**,**f**) under control and saline conditions (150 and 300 mM NaCl). Different letters indicate significant differences between treatments (*p* < 0.05).

**Figure 4 plants-11-00618-f004:**
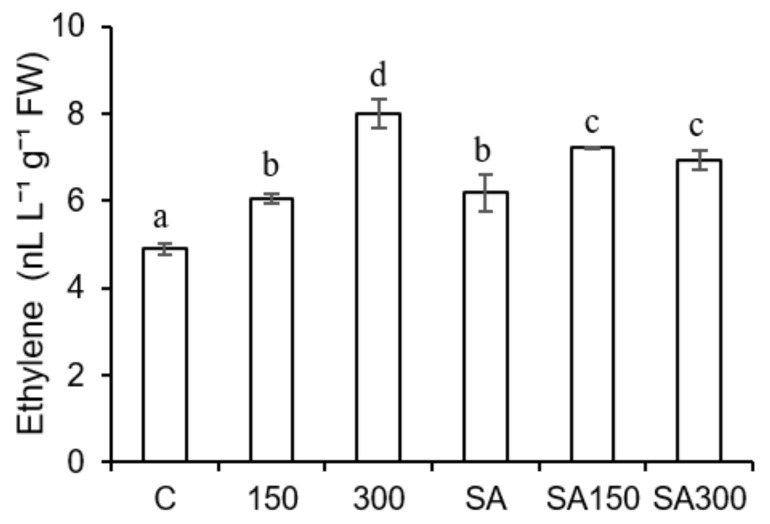
Effect of salicylic acid (SA) pre-treatment on ethylene levels of *Hordeum vulgare* L. ‘Ince-04′ plants under control and saline conditions (150 and 300 mM NaCl). Different letters indicate significant differences between treatments (*p* < 0.05).

**Figure 5 plants-11-00618-f005:**
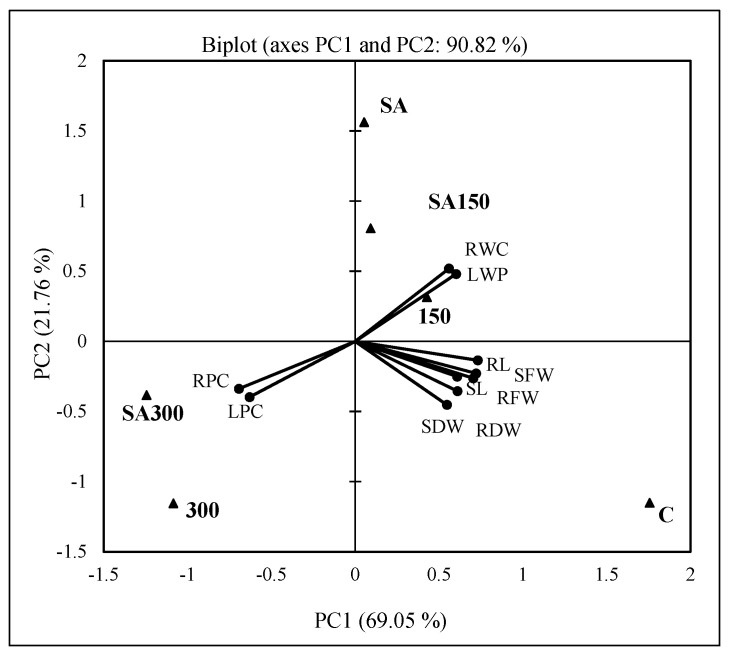
Bi−plot (PC1 × PC2) of PCA scores and loadings for shoot and root lengths (SL and RL), fresh and dry weights (SFW and SDW, RFW and RDW), leaf water potential (LWP) and relative water content (LRWC), and leaf and root proline contents (LPC and RPC). C; control, 150; 150 mM NaCl, 300; 300 mM NaCl, SA; 0.5 mM SA, SA150; 150 mM NaCl + 0.5 mM SA, SA300; 300 mM NaCl + 0.5 mM SA; RL, root length; SL, shoot length; RFW, root fresh weight; RDW, root dry weight; SDW, shoot dry weight; LRWC, leaf relative water content; LWP, leaf water potential; LPC, leaf proline content; RPC, root proline content. ● Variables and ▲ observations. Values in bold are different from 0 with a significance level of α = 0.05. Correlation matrix (Pearson, *r*) between growth parameters and stress treatments; RL → SL → RFW *r* = 0.926, 0.871, 0.973; SL → SFW *r* = 0.929; RFW → SFW-SDW *r* = 0.907, 0.952, 0.844; RDW → SDW *r* = 0.901; RWC → LWP → LPC → RPC *r* = 0.968, −0.898, −0.947; LWP → LPC−RPC *r* = −0.961, −0.970; LPC → RPC *r* = 0.939.

**Figure 6 plants-11-00618-f006:**
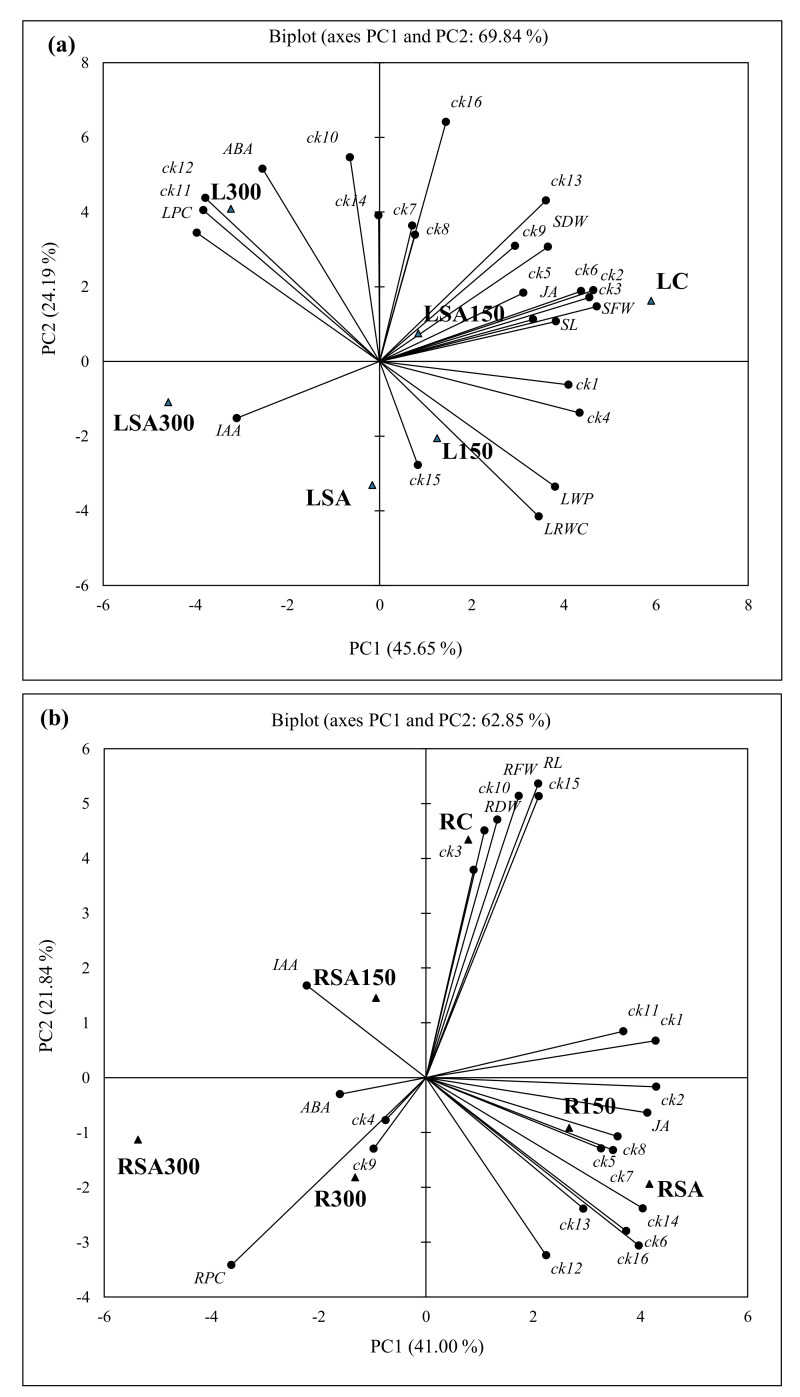
Bi−plot (PC1 x PC2) of scores and loadings for the PCA of all identified and quantified 15 cytokinins, IAA, ABA, and JA, and basic morphological and physiological parameters in leaves (**a**) and root (**b**) LC; leaf control, L150; leaf 150 mM NaCl, L300; leaf 300 mM NaCl, LSA; leaf 0.5 mM SA, LSA150; leaf 150 mM NaCl + 0.5 mM SA, LSA300; leaf 300 mM NaCl + 0.5 mM SA, RC; root control, R150; root 150 mM NaCl, R300; root 300 mM NaCI, RSA; root 0.5 mM SA, RSA150; root 150 mM NaCl + 0.5 mM SA, RSA300; root 300 mM NaCl + 0.5 mM SA. CKs: ck1; *t*Z, ck2; *t*ZR, ck3; *t*Z9G, ck4; *t*ZOG, ck5; *t*ZROG, ck6; *t*ZR′5MP, ck7; *c*Z, ck8; *c*ZR, ck9; *c*Z9G, ck10; *c*ZOG, ck11; *c*ZROG, ck12; *c*ZR′5MP, ck13; iP, ck14; iPR, ck15; iP9G, ck16; iZR′5MP. ● variables and ▲ observations. Values in bold are different from 0 with a significance level *α* = 0.05.

**Figure 7 plants-11-00618-f007:**
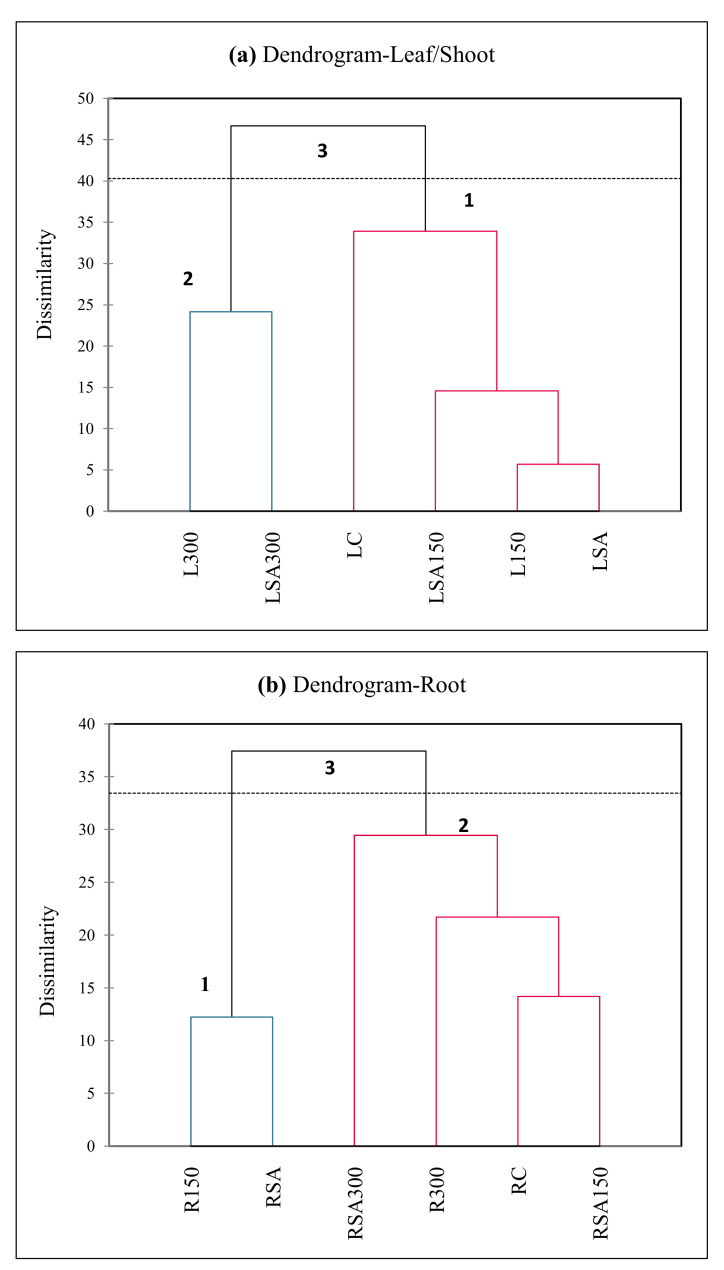
The clustering dendrograms representing dissimilarity between endogenous hormones and basic morphological and physiological parameters in leaves (**a**) and roots (**b**) in the barley seedlings under salt and exogenous SA treatments, obtained by the UPGMA clustering method with Euclidean distance, considering the concentration attained by the UPGMA clustering. LC; leaf control, L150; leaf 150 mM NaCl, L300; leaf 300 mM NaCl, LSA; leaf 0.5 mM SA, LSA150; leaf 150 mM NaCl + 0.5 mM SA, LSA300; leaf 300 mM NaCl + 0.5 mM SA, RC; root control, R150; root 150 mM NaCl, R300; root 300 mM NaCl, RSA; root 0.5 mM SA, RSA150; root 150 mM NaCl + 0.5 mM SA, RSA300; root 300 mM NaCl + 0.5 mM SA.

**Table 1 plants-11-00618-t001:** Effect of salicylic acid (SA) pre-treatment on growth and physiological parameters (length, fresh weight, dry weight, leaf relative water content, and leaf water potential) of *Hordeum vulgare* L. ‘Ince-04′ under saline conditions (150 and 300 mM NaCl). Values are mean ± SE (*n* = 6). The data with different letters in the same column at superscript are significantly different (*p* < 0.05).

Treatment	RL (cm)	SL (cm)	RFW (g)	SFW (g)	RDW (g)	SDW (g)	LRWC	LWP
C	45.0 ± 6.3 ^c^	46.4 ± 0.6 ^b^	1.82 ± 0.32 ^c^	3.02 ± 0.16 ^c^	0.17 ± 0.05 ^c^	0.39 ± 0.01 ^c^	87.4 ± 2.5 ^b^	−1.29 ± 0.09 ^a^
150	26.3 ± 3.7 ^ab^	40.0 ± 1.0 ^a^	1.26 ± 0.25 ^b^	1.78 ± 0.13 ^ab^	0.15 ± 0.02 ^bc^	0.32 ± 0.02 ^bc^	87.1 ± 2.1 ^b^	−1.28 ± 0.23 ^a^
300	22.3 ± 1.8 ^ab^	40.1 ± 5.1 ^a^	0.51 ± 0.15 ^a^	1.56 ± 0.34 ^ab^	0.11 ± 0.03 ^ab^	0.30 ± 0.10 ^bc^	77.7 ± 0.7 ^a^	−2.14 ± 0.18 ^b^
SA	29.0 ± 7.1 ^b^	41.7 ± 2.3 ^a^	0.62 ± 0.11 ^a^	1.68 ± 0.25 ^ab^	0.09 ± 0.00 ^a^	0.22 ± 0.06 ^ab^	91.2 ± 1.3 ^c^	−1.15 ± 0.21 ^a^
SA150	28.0 ± 3.1 ^b^	41.0 ± 1.2 ^a^	0.81 ± 0.19 ^a^	1.93 ± 0.41 ^b^	0.11 ± 0.02 ^ab^	0.24 ± 0.05 ^ab^	86.2 ± 2.5 ^b^	−1.24 ± 0.04 ^a^
SA300	19.0 ± 1.9 ^a^	40.8 ± 0.7 ^a^	0.49 ± 0.17 ^a^	1.43 ± 0.10 ^a^	0.10 ± 0.02 ^ab^	0.20 ± 0.01 ^a^	79.0 ± 2.5 ^a^	−2.11 ± 0.19 ^b^

C, control (0 mM NaCl and SA); SA, 0.5 mM salicylic acid; 150, 150 mM NaCl; 300, 300 mM NaCl; RL, root length; SL, shoot length; RFW, root fresh weight; SFW, shoot fresh weight; RDW, root dry weight; SDW, shoot dry weight; LRWC, leaf relative water content; LWP, leaf water potential.

**Table 2 plants-11-00618-t002:** Effect of salicylic acid (SA) pre-treatment on endogenous cytokinin levels (pmol g^−1^ FW) in *Hordeum vulgare* L. ‘Ince-04′ plants grown under control or saline conditions (150 and 300 mM NaCl). Values are mean ± SE (*n* = 6). The data with different letters in the same column at superscript are significantly different (*p* < 0.05).

		Bases	Ribosides	Nucleotides	*O*-Glucosides	9-Glucosides	Total CKs
Leaf							
	C	10.57 ± 1.81 ^b^	27.15 ± 5.38 ^b^	4.30 ± 0.47 ^b^	84.64 ± 15.9 ^bc^	9.01 ± 0.23 ^e^	135.67 ± 23.8 ^c^
	150	11.25 ± 0.19 ^b^	33.46 ± 1.23 ^c^	2.34 ± 0.05 ^a^	76.81 ± 1.42 ^abc^	6.98 ± 0.00 ^d^	130.84 ± 2.41 ^bc^
	300	15.33 ± 1.27 ^c^	59.21 ± 1.50 ^d^	8.10 ± 0.61 ^d^	105.08 ± 1.75 ^d^	6.15 ± 0.04 ^b^	193.87 ± 3.94 ^e^
	SA	9.90 ± 1.03 ^b^	28.64 ± 5.43 ^bc^	2.09 ± 0.18 ^a^	67.25 ± 3.49 ^a^	6.50 ± 0.22 ^c^	114.39 ± 10.4 ^ab^
	SA150	16.30 ± 0.25 ^c^	65.21 ± 1.25 ^e^	3.97 ± 0.06 ^b^	74.69 ± 4.12 ^ab^	7.01 ± 0.04 ^d^	167.18 ± 3.02 ^d^
	SA300	2.65 ± 0.04 ^a^	6.87 ± 0.86 ^a^	5.36 ± 0.29 ^c^	88.35 ± 2.93 ^c^	5.72 ± 0.02 ^a^	108.95 ± 4.10 ^a^
Root							
	C	2.02 ± 0.01 ^bc^	11.22 ± 1.03 ^c^	2.16 ± 0.53 ^a^	98.26 ± 4.78 ^d^	13.68 ± 0.71 ^cd^	127.33 ± 2.50 ^bc^
	150	2.33 ± 0.22 ^cd^	16.05 ± 0.12 ^d^	9.84 ± 2.25 ^d^	94.95 ± 1.48 ^cd^	6.19 ± 1.66 ^ab^	129.37 ± 2.33 ^bcd^
	300	2.29 ± 0.47 ^bcd^	16.34 ± 0.90 ^d^	8.36 ± 0.78 ^cd^	85.52 ± 4.73 ^b^	8.28 ± 2.39 ^abc^	120.80 ± 6.53 ^b^
	SA	2.60 ± 0.37 ^d^	20.13 ± 0.70 ^e^	6.91 ± 0.10 ^c^	91.24 ± 2.07 ^c^	10.54 ± 0.42 ^bc^	131.42 ± 3.66 ^cd^
	SA150	1.77 ± 0.10 ^b^	8.74 ± 0.20 ^b^	5.07 ± 0.62 ^b^	105.26 ± 1.42 ^e^	17.94 ± 6.85 ^d^	138.78 ± 9.19 ^d^
	SA300	0.62 ± 0.10 ^a^	1.31 ± 0.03 ^a^	1.53 ± 0.16 ^a^	49.94 ± 1.74 ^a^	4.46 ± 0.64 ^a^	57.85 ± 2.46 ^a^

C, control; SA, 0.5 mM salicylic acid; 150, 150 mM NaCl; 300, 300 mM NaCl; CK, cytokinin.

**Table 3 plants-11-00618-t003:** Effect of salicylic acid (SA) pre-treatment on total *cis*-zeatin (*c*Z), *trans*-zeatin (*t*Z), and isopentenyladenine (iP) types of *Hordeum vulgare* L. ‘Ince-04′ plant leaves and roots under control or saline conditions (150 and 300 mM NaCl). Values are mean ± SE (*n* = 6). The data with different letters in the same column at superscript are significantly different (*p* < 0.05).

		*c*Z-Types	*t*Z-Types	iP-Types
Leaf				
	C	106.21 ± 0.95 ^d^	16.61 ± 2.03 ^c^	12.85 ± 2.55 ^b^
	150	105.04 ± 0.35 ^c^	11.33 ± 1.83 ^ab^	14.47 ± 0.93 ^b^
	300	154.71 ± 0.30 ^f^	8.55 ± 3.02 ^a^	30.60 ± 0.61 ^c^
	SA	91.64 ± 0.10 ^a^	14.60 ± 2.06 ^bc^	13.15 ± 2.40 ^b^
	SA150	122.35 ± 0.42 ^e^	10.62 ± 3.79 ^ab^	34.21 ± 1.19 ^d^
	SA300	97.25 ± 0.14 ^b^	8.13 ± 3.50 ^a^	3.57 ± 0.73 ^a^
Root				
	C	105.86 ± 1.41 ^b^	13.16 ± 2.66 ^bc^	8.32 ± 1.26 ^c^
	150	111.53 ± 1.13 ^c^	7.05 ± 2.06 ^a^	10.79 ± 0.86 ^d^
	300	102.31 ± 2.32 ^b^	8.68 ± 4.35 ^ab^	9.81 ± 0.15 ^d^
	SA	103.98 ± 1.13 ^b^	12.38 ± 2.73 ^bc^	15.06 ± 0.20 ^e^
	SA150	114.90 ± 7.07 ^c^	17.19 ± 1.95 ^c^	6.69 ± 0.16 ^b^
	SA300	51.15 ± 0.49 ^a^	4.88 ± 1.92 ^a^	1.83 ± 0.05 ^a^

C, control; SA, 0.5 mM salicylic acid; 150, 150 mM NaCl; 300, 300 mM NaCl.

**Table 4 plants-11-00618-t004:** Loadings of the factor scores (*p* < 0.05) on principal components. ^A^ Leaf/shoot and ^B^ roots’ basic morphological and physiological parameters, cytokinin metabolites, IAA, BA, and JA measured in barley (*Hordeum vulgare* L.) seedlings under salt and exogenous salicylic acid conditions.

	Leaf/Shoot ^A^						Root ^B^				
Treatments *	F1	F2	F3	F4	F5	Treatments *	F1	F2	F3	F4	F5
LC	5.892	1.626	−2.473	0.500	−0.277	RC	0.790	4.343	−0.155	−0.970	1.230
L150	1.246	−2.060	1.823	1.534	1.296	R150	2.671	−0.917	−0.436	−2.217	−1.986
L300	−3.231	4.083	1.603	1.108	−0.378	R300	−1.321	−1.816	3.835	−0.522	0.958
LSA	−0.158	−3.309	1.149	0.073	−1.444	RSA	4.166	−1.938	−1.576	1.520	1.302
LSA150	0.839	0.754	1.478	−2.951	0.414	RSA150	−0.933	1.456	0.892	2.574	−1.780
LSA300	−4.587	−1.095	−3.580	−0.264	0.389	RSA300	−5.372	−1.127	−2.559	−0.385	0.277

* LC; leaf control, L150; leaf 150 mM NaCl, L300; leaf 300 mM NaCl, LSA; leaf 0.5 mM SA, LSA150; leaf 150 mM NaCl + 0.5 mM SA, LSA300; leaf 300 mM NaCl + 0.5 mM SA, RC; root control, R150; root 150 mM NaCl, R300; root 300 mM NaCl, RSA; root 0.5 mM SA, RSA150; root 150 mM NaCl + 0.5 mM SA, RSA300; root 300 mM NaCl + 0.5 mM SA.

## Data Availability

The data presented in the current study are available in the article and Appendix A.

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
