# Peer review of "The Effects of Exogenous Salicylic Acid on Endogenous Phytohormone Status in Hordeum vulgare L. under Salt Stress"

_plants, 2022, doi:10.3390/plants11050618_

Round 1

Reviewer 1 Report

This paper reports the effects of SA pre-treatment and salt stress on endogenous hormone contents in barley. The results bring novel insights into physiological state of SA-treated plants under salinity. Therefore, this MS is worth publishing in Plants. The manuscript, however, could be improved if the following issues are appropriately addressed.

  1. In Figure 1, scale bars should be in the same size to compare the length of each plant.

The leaves of all plants appear to be spirally wound. Are those their normal morphology?

  1. line160, I can not believe that SA150 is 40% higher than 150 in leaves.

  1. In Table 1, RWC should be ‘Leaf RWC’.

  1. I understand that SA-treatment usually make plants dehydrated by PCD or HR. So, it seems very unusual that RWC increased in SA-treated plants. Please explain this point.

Similarly, the combination of 0.5 mM SA pre-treatment and 300 mM NaCl seems to be severe condition for plants and may cause reduction of enzyme activities of many.

The levels of some hormones in 150 mM and 300 mM NaCl treated samples are extremely different. These may not be the effects of metabolic changes caused by SA pre-treatments, but rather the decrease in enzyme activity due to damage to plants caused by SA and salt stress.

  1. 5.1 and 2.5.2 should be unified because the authors compare SA-treated and SA-non-treated plants in 2.5.2. Make the subtitles of revised 2.5.1 and former 2.5.3 uniform. Similarly, make the subtitles of 2.5.4 and 2.5.5 (and the followings) uniform.

  1. English should be corrected throughout manuscript.

For example, the authors often use adverbs like adjectives, such as ‘moderately’.

In Table 2 and 3, (0 mM NaCl and SA) is not appropriate.

  1. Line538-541. This is not correct based on Table 1.

  1. The discussion is redundant and requires significant simplification, especially p13-14.

  1. Line 559. ‘However, it did non-significantly increase RWC under high salt stress, presumably because of osmotic changes resulting from high proline accumulation.’ This sentence is unclear. An increase in proline content should increase RWC by increasing osmotic pressure.

Line 588, cytokinins should be CKs.

Line 603-608, this part is not logically correct.

The contents of endogenous hormones due to salt stress after SA pre-treatment varied greatly depending on the salt concentration, which often contradicted the results of previous studies. Do the authors confirm the reproducibility of the results?

Also, the interpretations of those results in the discussion involve over-speculation.

  1. Please correct the notation of the bibliographic information in the References. Front Plant Sci has no pages.

  1. There is no explanation for PCA in the Discussion and Abstracts. Therefore, Fig6 may not be needed. At least, if it is left, it is necessary to eliminate the overlap of characters in Fig6C.

Author Response

Response to Reviewer 1 Comments

Manuscript ID: plants-1591790

Title: The Effects of Exogenous Salicylic Acid on Endogenous Phytohormone Status in Hordeum vulgare L. under Salt Stress

This paper reports the effects of SA pre-treatment and salt stress on endogenous hormone contents in barley. The results bring novel insights into physiological state of SA-treated plants under salinity. Therefore, this MS is worth publishing in Plants. The manuscript, however, could be improved if the following issues are appropriately addressed.

Thanks for reviewer’ comments for improving and encouraging our manuscript. We respond these comments with point-by-point.

Point 1: In Figure 1, scale bars should be in the same size to compare the length of each plant. The leaves of all plants appear to be spirally wound. Are those their normal morphology?

Response 1: Thank you for this valuable comment. Figure 1 was revised as mentioned. Barley plants were grown in an aerated hydroponic system containing Hoagland’s solution (Hoagland and Arnon, 1950), as described in M&M section. In another publication of Hoagland and Broyer (Hoagland, D.R. and Broyer, T.C. 1936. General nature of the process of salt accumulation by roots with description of experimental methods. Plant Physiology, 11(3), 471-507), they worked with barley plants grown under salinity conditions. In the light of these two publications, barley plants are grown in their best condition under salt stress. Therefore, the barley leaves curl as they grow. Namely, spirality of leaves are their normal morphology even in control conditions. You can see a sample barley photo below grown without any stress conditions.

Point 2: Line160, I can not believe that SA150 is 40% higher than 150 in leaves.

Response 2: We are really sorry of this mistake. It was revised as “…SA was 23.1% higher than…”. Besides, all manuscript was checked after this comment.

Point 3: In Table 1, RWC should be ‘Leaf RWC’.

Response 3: Thank you for this remark, sorry for this. It was revised both in Table 1 and also throughout the manuscript.

Point 4: I understand that SA-treatment usually make plants dehydrated by PCD or HR. So, it seems very unusual that RWC increased in SA-treated plants. Please explain this point.

Similarly, the combination of 0.5 mM SA pre-treatment and 300 mM NaCl seems to be severe condition for plants and may cause reduction of enzyme activities of many.

The levels of some hormones in 150 mM and 300 mM NaCl treated samples are extremely different. These may not be the effects of metabolic changes caused by SA pre-treatments, but rather the decrease in enzyme activity due to damage to plants caused by SA and salt stress.

Response 4: Thank you for your valuable comment. Most studies in literature reported positive effects of SA on plants grown under stress conditions, similar to our previous studies with other barley cultivars. However, one of the barley cultivar, Ince-04, displayed different responses to SA treatments. So, we want to represent these results with these conditions. On the other hand, most studies in the literature mentioned increasing in RWC with SA treatment which referred in our study. SA treatment increased leaf RWC in this barley cultivar might be the result of increasing proline content.

Point 5: 2.5.1 and 2.5.2 should be unified because the authors compare SA-treated and SA-non-treated plants in 2.5.2. Make the subtitles of revised 2.5.1 and former 2.5.3 uniform. Similarly, make the subtitles of 2.5.4 and 2.5.5 (and the followings) uniform.

Response 5: Thank you for this valuable comment. Subtitles were revised as mentioned.

Point 6: English should be corrected throughout manuscript.

For example, the authors often use adverbs like adjectives, such as ‘moderately’.

In Table 2 and 3, (0 mM NaCl and SA) is not appropriate.

Response 6: Thank you for the comment. Before submitted for the first time, English of MS was edited by Sees-editing Ltd., U.K. After this comment, the MS was revised by them to make it easier to read.

Besides, Table 2 and 3 were also revised as taking this into account.

Point 7: Line538-541. This is not correct based on Table 1.

Response 7: Thank you for this remark, sorry for this. Discussion part was revised towards this recommendation.

Point 8: The discussion is redundant and requires significant simplification, especially p13-14.

Response 8: Thank you for this comment. Discussion part was revised as mentioned.

Point 9: Line 559. ‘However, it did non-significantly increase RWC under high salt stress, presumably because of osmotic changes resulting from high proline accumulation.’ This sentence is unclear. An increase in proline content should increase RWC by increasing osmotic pressure.

Line 588, cytokinins should be CKs.

Line 603-608, this part is not logically correct.

The contents of endogenous hormones due to salt stress after SA pre-treatment varied greatly depending on the salt concentration, which often contradicted the results of previous studies. Do the authors confirm the reproducibility of the results?

 Also, the interpretations of those results in the discussion involve over-speculation.

Response 9: Line 559. ‘However, it did non-significantly increase RWC under high salt stress, presumably because of osmotic changes resulting from high proline accumulation.’ This sentence is unclear. An increase in proline content should increase RWC by increasing osmotic pressure.

Proline could be more important in inducing osmotic adjustment in plant submitted to salinity stress. However, apart from proline, organic and/or inorganic solutes such as ions, amino acids, organic acids etc. involve in osmotic adjustment to maintain the turgor. In current study, SA treatment did not show signigicant effects on RWC while proline accumulated under these conditions, as compared to non-SA-treated salt-stressed barley cultivar. This cultivar altered proline content to adapt osmotic stress and maintain water content. This result suggest that this cultivar has different salt sensitivity from those other barley cultivars. In the literature, studies reported that salt-tolerant plants showed less reduction in RWC than salt-sensitive plants during salinity stress. Moreover, salt-tolerant cultivars gradually accumulates proline content, and its osmotic potentials of the salt-stressed plants were not significantly different from those of their non-stressed control counterparts. Hence, this barley cultivar might not be used the osmotic adjustment as a predominant mechanism to cope with salinity stress.

Line 588, cytokinins should be CKs.

Also, abreviation of CKs was used instead of cytokinins.

Line 603-608, this part is not logically correct.

Pardon! Our mistake. It was improved accordingly!

SA pre-treatment under non-saline conditions increased IAA levels in roots but not in leaves. Similarly, SA pre-treatment had no significant effect on IAA levels in the leaves of barley seedlings under moderate salt stress (150 mM NaCl) but increased them in roots compared to 150 controls. Conversely, the SA + 300 mM NaCl treatment dramatically increased IAA levels in leaves but had no significant effect on those in roots. However, a study on wheat revealed that SA treatment prior to sowing prevented an NaCl-induced decline in IAA levels of 2% [11].

The contents of endogenous hormones due to salt stress after SA pre-treatment varied greatly depending on the salt concentration, which often contradicted the results of previous studies. Do the authors confirm the reproducibility of the results?

There is very precise approach for phytohormone analysis which we use. The phytohormones are always quantified by the standard isotope-dilution method (internal standardization of all metabolites) using three technical replicates per biological sample. Stable isotope-labelled internal standards were used as references to quantify levels of the target analytes. Furthermore, two biological replicates with these three technical replicates each (n = 6) were performed for each experiment. The samples were collected from two independent biological experiments. These analytical results are really very correct. We think that it shuld be more realated to some differences between the experiments.

Also, the interpretations of those results in the discussion involve over-speculation.

It is not clear which type of results. We imporved IAA discussion as presented above.

Point 10: Please correct the notation of the bibliographic information in the References. Front Plant Sci has no pages.

Response 10: Many thanks for this comment and we are very really sorry of this. Page numbers were eliminated.

Point 11: There is no explanation for PCA in the Discussion and Abstracts. Therefore, Fig6 may not be needed. At least, if it is left, it is necessary to eliminate the overlap of characters in Fig6C.

Response 11: Thank you for your valuable comment. All PCA analyses were redone and figures were revised. Besides, cluster analysis was performed.

Reviewer 2 Report

The manuscript deals with a topic of interest and includes a large dataset. There are very few publications on the effects of exogenous salicylic acid on endogenous phytohormones. The manuscript is also well written.

However, there is a major flaw in relation to the working hypothesis and the results obtained. The interest of this study is that the application of exogenous SA can alleviate the adverse effects of stress, but the analysis of growth parameters clearly indicated the opposite. Not only did SA not alleviate salinity, but it has a detrimental effect on plant growth even in the absence of salt. Moreover, the growth of plants subjected to both salt and SA treatments was reduced more than in plants treated only with saline solutions.  Therefore, there is no way to detect a positive effect of SA under these experimental conditions, despite the author's efforts to justify it. In my opinion the study is interesting for its novelty, but the authors' conclusions do not correspond to the results obtained. I suggest that the authors rewrite the manuscript and stick to the results, without trying to justify a positive effect of SA, which is non-existent in this work.

Another suggestion is that the Principal Component Analysis should include all the parameters analysed. The growth parameters are essential for the interpretation of the results. A more appropriate presentation would be the separation of the loading plot of the principal components by that of the scatter plots of the PCA scores. Another possibility is to use a hierarchical cluster analysis and a heat map.  This would make the interpretation of the multivariate analysis more concise and clarifying.

Author Response

Response to Reviewer 2 Comments

Manuscript ID: plants-1591790

Title: The Effects of Exogenous Salicylic Acid on Endogenous Phytohormone Status in Hordeum vulgare L. under Salt Stress

The manuscript deals with a topic of interest and includes a large dataset. There are very few publications on the effects of exogenous salicylic acid on endogenous phytohormones. The manuscript is also well written.

Thanks for reviewer’ comments for improving and encouraging our manuscript. We respond these comments with point-by-point.

Point 1: However, there is a major flaw in relation to the working hypothesis and the results obtained. The interest of this study is that the application of exogenous SA can alleviate the adverse effects of stress, but the analysis of growth parameters clearly indicated the opposite. Not only did SA not alleviate salinity, but it has a detrimental effect on plant growth even in the absence of salt. Moreover, the growth of plants subjected to both salt and SA treatments was reduced more than in plants treated only with saline solutions.  Therefore, there is no way to detect a positive effect of SA under these experimental conditions, despite the author's efforts to justify it. In my opinion the study is interesting for its novelty, but the authors' conclusions do not correspond to the results obtained. I suggest that the authors rewrite the manuscript and stick to the results, without trying to justify a positive effect of SA, which is non-existent in this work.

Response 1: Thank you for your valuable comments. The manuscript was revised on a large scale according to comments.   

Point 2: Another suggestion is that the Principal Component Analysis should include all the parameters analysed. The growth parameters are essential for the interpretation of the results. A more appropriate presentation would be the separation of the loading plot of the principal components by that of the scatter plots of the PCA scores. Another possibility is to use a hierarchical cluster analysis and a heat map.  This would make the interpretation of the multivariate analysis more concise and clarifying.

Response 2: Thank you for your valuable comment. All PCA analyses were redone and figures were revised. Besides, cluster analysis was performed.

Reviewer 3 Report

The review of the paper by Torun et al. for PLANTS MDPI.

In the introduction section there is lack of link between particular paragraphs. I suggest also to modified the aim of the work, as use of methods should not be the main aim of the study. We choose the methods with is the best to realize the aim (L88).

Figure 1 – it is difficult to judge about the differences based on this picture. You have presented only one plant per each treatments. If you have picture which visualize the changes described by you please replace it.

I have some doubts about SA concentration used in this study. When I look on the results presented by you it seems for me that plants pre-treated just with SA and then used as a control exhibited worse physiological condition. Most of the parameters showed in Table 1 decreased. Such results suggest that this concentration of SA is cytotoxic for plants and have inhibitory effect on plant growth and development. In L538-9 you have stated that “SA treatment also had non-significant changes in the growth of barley shoots”. According to Table 1 it is not true, as f.eg. SL, SFW, SDW decreased in shoots from plants pre-treated with SA, when compared to the Control treatment.

I ask you to verify he concentration of sodium hypochloride. Did you used 5% sodium hypochloride or 5% bleach, which is usually some solution (mostly 5%) of sodium hypochloride. In my lab we used max. 20% bleach which stated for 1% sodium hypochloride. However it strongly depends from species used in the study.

L713 please correct indexes

L716-717 I am not sure as most od the treatments after 24h pretreatment were exposed for 4 days for salt stress. Do you exposed SA pre-treatment plants for 4 days in water or used directly after. L721-722 suggested that you grew the plants for 4 days, but it is not stated before.

Author Response

Response to Reviewer 3 Comments

Manuscript ID: plants-1591790

Title: The Effects of Exogenous Salicylic Acid on Endogenous Phytohormone Status in Hordeum vulgare L. under Salt Stress

The review of the paper by Torun et al. for PLANTS MDPI.

Thanks for reviewer’ comments for improving and encouraging our manuscript. We respond these comments with point-by-point.

Point 1: In the introduction section there is lack of link between particular paragraphs. I suggest also to modified the aim of the work, as use of methods should not be the main aim of the study. We choose the methods with is the best to realize the aim (L88).

Response 1: Thank you for this valuable comment. Introduction part was revised as mentioned and paragraphs were reorganized. Sorry for using of methods in the introduction part. Hence, the aim of work was revised as taking this into account.

Point 2: Figure 1 – it is difficult to judge about the differences based on this picture. You have presented only one plant per each treatments. If you have picture which visualize the changes described by you please replace it.

Response 2: Many thanks for this valuable comment. Figure 1 was revised the new one as requested.

Point 3: I have some doubts about SA concentration used in this study. When I look on the results presented by you it seems for me that plants pre-treated just with SA and then used as a control exhibited worse physiological condition. Most of the parameters showed in Table 1 decreased. Such results suggest that this concentration of SA is cytotoxic for plants and have inhibitory effect on plant growth and development. In L538-9 you have stated that “SA treatment also had non-significant changes in the growth of barley shoots”. According to Table 1 it is not true, as f.eg. SL, SFW, SDW decreased in shoots from plants pre-treated with SA, when compared to the Control treatment.

Response 3: The selection of the dose used of SA based on previous researches (EI-Tayeb, 2005; Ma et al., 2017; Farhangi-Abriz and Ghassemi-Golezani, 2018; Es-sbihi, et al., 2021; etc.) and our preliminary experiments and papers (Torun et al., 2020). We have worked with barley cultivars for many years. Based on our preliminary experiments, SA concentration of 0.5 mM was chosen that measured the ability of various SA concentrations (0.1, 0.5 and 1.0 mM) to induce radicle growth and rescue seedlings from the detrimental effects of NaCl. However, this species used in this study (Ince-04) showed different respoenses to SA. Hence, we want to show these differences in the MS.

Sorry for the presentation of these results and discussion parts. We revised the MS as reviewer’s valuable comments. 

EI-Tayeb, M.A. Response of barley grains to the interactive effect of salinity and salicylic acid. Plant Growth Regul. 2005, 4, 215-224.

Ma, X.; Zheng, J.; Zhang, X.; Hu, Q.; Qian, R. Salicylic acid alleviates the adverse effects of salt stress on Dianthus superbus (Caryophyllaceae) by activating photosynthesis, protecting morphological structure, and enhancing the antioxidant system. Front. Plant Sci. 2017, 8, (600).

Farhangi-Abriz, S., & Ghassemi-Golezani, K. (2018). How can salicylic acid and jasmonic acid mitigate salt toxicity in soybean plants?. Ecotoxicology and Environmental Safety, 147, 1010–1016.

Es-sbihi, F.Z., Hazzoumi, Z., Aasfar, A. et al. Improving salinity tolerance in Salvia officinalis L. by foliar application of salicylic acid. Chem. Biol. Technol. Agric. 8, 25 (2021).

Torun, H.; Novák, O.; Mikulík, J.; Pěnčík, A.; Strnad, M.; Ayaz, F.A. Timing-dependent effects of salicylic acid treatment on phytohormonal changes, ROS regulation, and antioxidant defense in salinized barley (Hordeum vulgare L.). Sci. Rep. 2020, 10, 13886.

Point 4: I ask you to verify he concentration of sodium hypochloride. Did you used 5% sodium hypochloride or 5% bleach, which is usually some solution (mostly 5%) of sodium hypochloride. In my lab we used max. 20% bleach which stated for 1% sodium hypochloride. However it strongly depends from species used in the study.

Response 4: Thank you for this comment and it is a good remark. We used 5% sodium hypochlorite for sterilization of barley seeds. It was also revised in M&M section.

Point 5: L713 please correct indexes

Response 5: Sorry for this mistake. They were revised.

Point 6: L716-717 I am not sure as most od the treatments after 24h pretreatment were exposed for 4 days for salt stress. Do you exposed SA pre-treatment plants for 4 days in water or used directly after. L721-722 suggested that you grew the plants for 4 days, but it is not stated before.

Response 6: We are really sorry of this confusion. Plants were grown in aerated Hoagland solution for 16 days and the solution was refreshed every two days. After 16 days, SA applied to plants (expressed as SA pre-treatment) for 24 h in Hoagland solution. For salinity treatments, Hoagland was renewed and NaCl was dissolved in Hoagland solution. After 24 h SA-pretreatments, the seedlings were exposed to 150 or 300 mM NaCl added to the nutrient solution for 4 days. At the end of the 4 days NaCl treatment period, the barley plants were harvested. We were adding appropriate wording in this sentence in M&M section and revised it as shown below.

“After 16 days under these conditions, pots were randomly divided into six experimental groups: C: untreated control plants, 150: plants treated with 150 mM NaCl to induce moderate salt stress, 300: plants treated with 300 mM NaCl to induce high salt stress, SA: plants treated with 0.5 mM SA for 24 h (0 mM NaCl), SA150: plants preincubated for 24 h with 0.5 mM SA then cultivated for 4 days in 150 mM NaCl, SA300: 0.5 mM SA pretreatment for 24 h followed by growth in 300 mM NaCl for 4 days. SA and/or NaCl were applied to the plants in a Hoagland nutrient solution. There were 2 replicates per treatment and 16 plants per replicate. At the end of the 4 days NaCl treatment period, the barley plants were harvested. Roots and shoots were harvested separately from all groups of seedlings and stored at -80 °C for further analysis.”

Round 2

Reviewer 2 Report

I consider the revised version of the manuscript acceptable for publication. The authors followed my recommendations. The multivariate analysis has been improved and the new PCA graphs are much more explanatory. 

Reviewer 3 Report

The authors responded all my questions and provided corresponding modification in the text. The new version of the manuscript is prepared in acceptable form, so I do not have more objectives against publication of this paper in Plants.